# Application of the City Blueprint Approach in Landlocked Asian Countries: A Case Study of Ulaanbaatar, Mongolia

**Enkhuur Munkhsuld [1], Altansukh Ochir [1,*], Steven Koop [2,3], Kees van Leeuwen [2,3] and Taivanbat Batbold [1]**

[1] School of Engineering and Applied Sciences, National University of Mongolia, Sukhbaatar district, Ulaanbaatar 14201, Mongolia; altansukh@seas.num.edu.mn (A.O.); taivnaa.etko@gmail.com (T.B.)

[2] KWR Water Research Institute, Groningenhaven 7, 3433 PE Nieuwegein, The Netherlands; stef.koop@kwrwater.nl (S.K.); kees.van.leeuwen@kwrwater.nl (K.v.L)

[3] Copernicus Institute of Sustainable Development, Utrecht University, Princetonlaan 8a, 3508 TC Utrecht, the Netherlands

* Correspondence: altansukh@seas.num.edu.mn

**Abstract:** Urbanization is a major global development. At present, more than half of the world population lives in urban areas, i.e., cities. One of the fundamental requirements of citizens is safe and sufficient drinking water. The premises for water security are adequate water management and governance. In this study, we determine priorities for Integrated Water Resources Management (IWRM) and assess the governance capacities of different organizations to address IWRM in Ulaanbaatar, the capital of the landlocked Asian country Mongolia. We apply the City Blueprint Approach (CBA), a diagnosis tool, to assess IWRM in Ulaanbaatar city, Mongolia. The overall score, the Blue City Index (BCI), is 2.3 points for Ulaanbaatar, which categorizes the city as wasteful. Flood risk and economic pressure have a great impact on the water sector in Ulaanbaatar city. In particular, Ulaanbaatar's waste water treatment (WWT) can be improved. Often, only primary and a small portion of secondary WWT is applied, leading to large-scale pollution. Water consumption and infrastructure leakages are high due to the lack of environmental awareness and infrastructure maintenance. Operation cost recovery is not sufficient to sustain urban water services in Ulaanbaatar. Water governance and more specifically monitoring, evaluation and statutory compliance are among the factors that need to be addressed.

**Keywords:** city blueprint approach; integrated water resources management; water governance; landlocked country; Mongolia

## 1. Introduction

Across the globe, water management faces many challenges such as population growth, urbanization and climate change [1]. Water infrastructure is also very expensive. A recent report published by the Organisation for Economic Co-operation and Development (OECD) [2] summarizes the challenges related to the UN Sustainable Development Goal 6 (SDG 6) of good water and sanitation for all. More than two billion people do not have access to good drinking water, more than four billion people are deprived of adequate sanitation, and the projected cost for water infrastructure up to 2050 exceeds 22 trillion USD.

Although cities are vulnerable, they also have the innovative capacity to address these challenges. Accordingly, urban communities may be better engaged in instigating local action to address global challenges [3]. Drinking water supply and waste water treatment facilities are crucial

factors to sustain public health. It is estimated that by 2050, approximately 68% of the world population will live in urban areas [4]. This increase is driven by high fertility in sub-Saharan Africa, whose population is forecasted to more than double in the next 40 years, and by a modest 23% growth of Asia's huge population [5]. Out of 25 megacities listed as the "most densely populated", 17 are located in Asia, including Manila, Dhaka and Jakarta [6]. Therefore, in-depth assessment of cities is essential to provide comprehensive solutions to these urban challenges and fulfil the international ambitions related to clean water and sanitation as formulated in SDG 6 [7–9].

The water sector in Mongolia has developed scientifically since 1900. In fact, Mongolia's urban water research and management is a relatively new field, following developments in other sectors such as agricultural water supply, industrial water supply and hydrological research in general [10]. Based on a project entitled "Strengthening water resources management in Mongolia", implemented between 2007 and 2012, the Government of Mongolia decided to implement Integrated Water Resources Management (IWRM) at river basin level. It did so by developing resolution #332 in 2009 to divide the whole territory of Mongolia into 29 larger river basins (Figure 1). Since then, river basin authorities have been created and IWRM of river basins was implemented together with IWRM for the entire country in 2013 [11]. The Mongolian law on water was approved by the Government of Mongolia in 2012. According to article 17.1.2 of the Mongolian law on water, the River Basin Authority is responsible for improvement and implementation of IWRM at river basin level, and according to article 20.1, the River Basin Council provides recommendations and options for the improvement and implementation of river basin management plans. Moreover, the National Program on Water (NWP) was approved by the Government of Mongolia in 2011 and now includes IWRM of river basins. Ulaanbaatar was mentioned as one of the provisions of the NWP, not as a separate plan for Ulaanbaatar city.

Responsibilities for water are divided across a variety of organizations such as (a) the Ministry of Environment and Tourism (water policy and management), (b) the Ministry of Food, Agriculture and Light Industry (irrigation and water supply of food industry), (c) the Ministry of Education, Culture, Science and Sports (research policy), (d) the Ministry of Energy (hydropower policy), (e) the Ministry of Health (public health and water quality standards), (f) the Ministry of Construction and Urban Development (water and sanitation services to households and industrial users—piped systems, tanks and pumps), (g) the Ministry of Finance (investment and water fees), and (h) a number of other water-related governmental organizations [12]. The current institutional fragmentation hinders the development of inter-sectorial perspectives and science–policy interactions, both at the national and local level.

Based on open-source information about doctoral studies on water research in Mongolia, a total of 111 studies have been done since 1960: irrigation and hydro-technology represent 38% (42), hydrobiology and hydro-ecology 15.5% (17), hydrology 15.5% (17), hydrochemistry and water hygiene 14% (16), water supply and treatment 11% (12), and only 6% (7) are for water economy and management [10].

In the context of water governance fragmentation, there is limited available knowledge on water management,, and because Mongolia is a country with scarce and unevenly distributed water resources [12], our research has three complementary objectives: (I) to describe current integrated water management and governance practices in a landlocked Asian country, i.e. Mongolia; (II) to assess urban water management of the capital, i.e., Ulaanbaatar city; and iii) to review the City Blueprint Approach for its applicability in the context of Mongolia and other Asian landlocked countries. The present study is the first assessment of IWRM at the city level in Mongolia.

## 2. Materials and Methods

### 2.1. The City Blueprint Approach

This study was carried out following the City Blueprint Approach (CBA) developed by KWR Water Research Institute in the Netherlands to determine the main challenges, to highlight critical aspects and to give recommendations based on the analysis of Ulaanbaatar, representing an Asian

city in a landlocked country. The CBA is a diagnosis tool for urban water management and consists of three sub-frameworks. The main challenges of cities are assessed with the Trends and Pressures Framework (TPF). How cities are managing their water systems is assessed with the City Blueprint Framework (CBF). Where cities can improve their water governance is assessed with the Governance Capacity Framework (GCF) [13–15]. CBA is one of the actions of the European Innovation Partnership of Water of the European Commission. By the end of 2019, the CBA has been used in about 80 municipalities in more than 40 countries [16]. The work includes 11 Asian countries and only one landlocked country, Hungary, where the capital Budapest was assessed.

### 2.1.1. The Trends and Pressures Framework (TPF)

The TPF consists of 12 descriptive indicators to summarize the exogenous social, environmental and financial conditions within which water managers have to operate (Table 1). Each indicator is scaled from 0 to 4 points, where a higher score represents a greater pressure or concern [13,14]. Most scores of the indicators are calculated based on international data sources, for example, the World Bank, World Health Organization and Food and Agricultural Organization. Details of the indicators, data sources and sample calculations are given in the E-Brochure [17].

**Table 1.** Indicators of the Trends and Pressures Framework and data sources [17].

| Categories | Indicators | Data Sources |
| --- | --- | --- |
| Social pressures | 1. Urbanization rate | CIA: The World Factbook |
| | 2. Burden of disease | WHO |
| | 3. Education rate | World Bank |
| | 4. Political instability | World Bank |
| Environmental pressures | 5. Flooding | EEA, local data source |
| | 6. Water scarcity | IGRAC [1], WRI [2], OECD [3], EEA [4] |
| | 7. Water quality | EIP [5], EEA [4] |
| | 8. Heat risk | EEA [4] |
| Financial pressures | 9. Economic pressure | IMF [6] |
| | 10. Unemployment rate | World Bank |
| | 11. Poverty rate | World Bank |
| | 12. Inflation rate | World Bank |

[1] International Groundwater Resources Assessment Centre [2] World Resources Institute [3] Organisation for Economic Co-operation and Development [4] European Environment Agency [5] European Innovation Partnerships. [6] International Monetary Fund

### 2.1.2 The City Blueprint Framework

The CBF is a baseline assessment or quick scan that evaluates the actual state of a city's IWRM and shows the scores of the indicators in a spider diagram [13–15]. The result of this assessment is the first step in the strategic planning process of IWRM in cities [18]. The CBF consists of 25 indicators divided into seven broad general categories (Table 2). All 25 indicators are scored from 0 (low performance) to 10 (high performance). The CBF provides a thorough understanding of the main challenges and may assist in prioritizing IWRM management options. The geometric mean of these indicators is the Blue City Index (BCI) [13,14]. Details of the indicators, data sources and sample calculations are given in the E-Brochure [17].

**Table 2.** Indicators of the City Blueprint Framework and data sources [17].

| Categories | Indicators | Data Sources |
| --- | --- | --- |
| Water quality | 1. Secondary waste water treatment (WWT) | IWA [1] Water Wiki |
| | 2. Tertiary WWT | IWA Water Wiki |
| | 3. Groundwater quality | EEA or local data sources |
| Solid waste treatment | 4. Solid waste collected | OECD: Environment at a glance |
| | 5. Solid waste recycled | OECD: Environment at a glance |
| | 6. Solid waste energy recovered | OECD: Environment at a glance |

| Basic water services | 7. Access to drinking water | WHO/UNICEF |
| | 8. Access to sanitation | WHO/UNICEF |
| | 9. Drinking water quality | WHO/UNICEF |
| Waste water treatment | 10. Nutrient recovery | OECD: Environment at a glance |
| | 11. Energy recovery | OECD: Environment at a glance |
| | 12. Sewage sludge recycling | OECD: Environment at a glance |
| | 13. WWT energy efficiency | Local data sources |
| Infrastructure | 14. Stormwater separation | Local data sources |
| | 15. Average age of sewer | Local data sources |
| | 16. Water system leakages | Green City Index reports |
| | 17. Operation cost recovery | IBNET |
| Climate robustness | 18. Green space | EEA |
| | 19. Climate adaptation | UNEP |
| | 20. Drinking water consumption | Local data sources |
| | 21. Climate-robust buildings | Local data sources |
| Governance | 22. Management and action plans | Local data sources |
| | 23. Public participation | World Bank |
| | 24. Water efficiency measures | Local data sources |
| | 25. Attractiveness | Local data sources |
| Overall score | Blue City Index, the geometric mean of 25 indicators varying from 0 to 10 | |

[1] International Water Association.

The CBF uses scientific literature, websites and official reports, preferably at city level, such as WHO, UNEP, World Bank as its source for calculating or assigning a score for each indicator [17]. Based on the BCI and similarities in the indicators' scores, cities are categorized into the following five categories provided in Table 3 [14,17].

**Table 3.** Categorization of different levels of IWRM [14,17].

| BCI Score | Categorization of IWRM in Cities |
| --- | --- |
| 0–2 | 'Cities lacking basic water services'<br>Access to potable drinking water of sufficient quality and access to sanitation facilities are insufficient. Typically, water pollution is high due to a lack of waste water treatment (WWT). Solid waste production is relatively low but is only partially collected and, if collected, almost exclusively put in landfills. Basic water services cannot be expanded or improved due to rapid urbanization. Improvements are hindered due to governance capacity and funding gaps. Deficient funding and governance capacity are a reason for being stuck in terms of basic water services. |
| 2–4 | 'Wasteful cities'<br>Basic water services are largely covered, while WWT is poorly covered. Often, only primary and a small portion of secondary WWT is applied, leading to large-scale pollution. Water consumption and infrastructure leakages are high due to the lack of environmental awareness and infrastructure maintenance. Solid waste production is high, and waste is almost completely dumped in landfills. Governance is reactive, and community involvement is low. |
| 4–6 | 'Water efficient cities'<br>Cities implementing centralized, well-known, technological solutions to increase water efficiency and to control pollution. Secondary WWT coverage is high, and the share of tertiary WWT is rising. Water-efficient technologies are partially applied; infrastructure leakages are substantially reduced, but water consumption is still high. Energy recovery from WWT is relatively high, while nutrient recovery is limited. Both solid waste recycling and energy recovery are partially applied. These cities are often vulnerable to climate change due to poor adaptation strategies, limited stormwater separation and low green surface ratios. Governance and community involvement have improved. |
| 6–8 | 'Resource efficient and adaptive cities'<br>WWT techniques to recover energy and nutrients are often applied. Solid waste recycling and energy recovery are largely covered, whereas solid waste production has not yet been reduced. |

| | |
|---|---|
| | Water-efficient techniques are widely applied, and water consumption has been reduced. Climate adaptation in urban planning is applied. Integrative, centralized and decentralized planning as well as long-term planning, community involvement and sustainability initiatives are established to cope with limited resources and climate change. |
| 8–10 | 'Water wise cities'<br>No city has scored in this category, yet. These cities apply full resource and energy recovery in their WWT and solid waste treatment, fully integrate water into urban planning, have multi-functional and adaptive infrastructures, and local communities promote sustainable integrated decision-making and behavior. Cities are largely water self-sufficient, attractive, innovative and circular by applying multiple (de)centralized solutions. |

Reports on the assessment of cities in the USA and Asia using the CBA, including the GCF for some cities, have been published recently [19,20].

### 2.1.3. The Governance Capacity Framework (GCF)

The governance capacity was analyzed to address these challenges with the GCF by interviewing water-related stakeholders (government institutions, NGOs, universities, research agents, authorities).

The GCF analyzes the governance capacity of a city to address a specific common water challenge. The first city in which this assessment has been performed was the city of Amsterdam [21]. The GCF provides a broad view on the capacities of urban stakeholders to address common water challenges. The GCF is a standardized methodology to assess governance-related aspects (Table 4) on, e.g., IWRM, water scarcity, flood risk, waste water treatment, solid waste treatment, urban heat islands and/or water reuse. This is done by performing semi-structured interviews with stakeholders from corresponding authorities, government bodies, and researchers to assess the real situation. Recently published examples of integrated analyses of cities, including assessment of governance capacities, are Seoul [22] and Cape Town [23]. The GCF is structured into three dimensions (knowing, wanting and enabling), nine key conditions and 27 indicators (Table 4). A Likert-type scaling is used to give scores on each indicator, which range from very encouraging (++) to very limiting (− −) [21–23].

**Table 4.** Indicators of the Governance Capacity Framework [21].

| Conditions | Indicators |
|---|---|
| 1 Awareness | 1.1 Community knowledge<br>1.2 Local sense of urgency<br>1.3 Behavioral internalization |
| 2 Useful knowledge | 2.1 Information availability<br>2.2 Information transparency<br>2.3 Knowledge cohesion |
| 3 Continuous learning | 3.1 Smart monitoring<br>3.2 Evaluation<br>3.3 Cross-stakeholder learning |
| 4 Stakeholder engagement process | 4.1 Stakeholder inclusiveness<br>4.2 Protection of core values<br>4.3 Progress and variety of options |
| 5 Management ambition | 5.1 Ambitious and realistic management<br>5.2 Discourse embedding<br>5.3 Management cohesion |
| 6 Agents of change | 6.1 Entrepreneurial agents<br>6.2 Collaborative agents<br>6.3 Visionary agents |
| 7 Multi-level network potential | 7.1 Room to maneuver<br>7.2 Clear division of responsibilities<br>7.3 Authority |

| 8 Financial viability | 8.1 Affordability |
| | 8.2 Consumer willingness to pay |
| | 8.3 Financial continuation |
| 9 Implementation capacity | 9.1 Policy instruments |
| | 9.2 Statutory compliance |
| | 9.3 Preparedness |

Details of the indicators, data sources and sample calculations are given in the E-Brochure [17].

*2.2. Study Area*

Mongolia is one of the 49 landlocked countries in the world, fully surrounded by land [24]. As of 2018, 46% (1,491,375 out of 3,238,479) of the whole population of Mongolia lives in Ulaanbaatar [25]. The total territory occupies 1,553,560 square km, and the population density is 1.9 person/square km [26]. Ulaanbaatar, the largest city and capital of Mongolia, is developing rapidly, both in terms of economic and population growth. Unfortunately, population settlements and industrial activities have intensified over the past two decades, leading to increased water consumption and adverse environmental effects such as depletion and pollution of the water resources [27]. The Ministry of Nature, Environment and Tourism of Mongolia implemented decision #332 in 2009 on river basin areas, and accordingly, the whole territory of Mongolia was divided into 29 larger river basins (Figure 1). This is being used to improve IWRM in Mongolia [27]. According to this decision, there are 29 water basin areas and Ulaanbaatar city falls within the Tuul River basin (49,416 square km).

The data used in all calculations were collected from the Mongolian statistical information service, Water Supply and Sewerage Authority in Ulaanbaatar, national and international reports, scientific papers, textbooks and official websites. Details of TPF, CBF and GCF methods are provided in the Supplementary Materials. Overall, the key information sources that have been consulted for the TPF and CBF indicators are listed in Table 1 and Table 2. In this GCF assessment of Ulaanbaatar, a total of 15 people participated in the interviews based on the GCF questionnaire to assess the current situation. People involved were a water security engineer at the Water Supply and Sewerage Authority in Ulaanbaatar, the head of the Hydrology sector at the Information and Research Institute of Meteorology, Hydrology and Environment, a professor at the National University of Mongolia, a professor at the Mongolian University of Science and Technology, two specialists from the Land Management and Water Policy Coordination department at the Ministry of Environment and Tourism, and two specialists from the Basin Authorization department at the Ministry of Environment and Tourism. As explained, these fifteen interviewees represented different stakeholders responsible for different water issues such as policy development, policy implementation, and scientific research. Moreover, all 15 stakeholders had been working in the water sector for more than nine years. During the interviews, a broad range of questions, including drinking water, waste water, water users (residents of the city), water supply, legal aspects and management practices were addressed. Based on a Likert-type scoring of each of the 27 indicators (Table 4) using a 5-point scale: very encouraging (++), encouraging (+), indifferent (0), limiting (−), and very limiting (−−) [21], a clear overview was obtained about the water-related governance capacities in Ulaanbaatar, specifically on IWRM, as shown in Table 5.

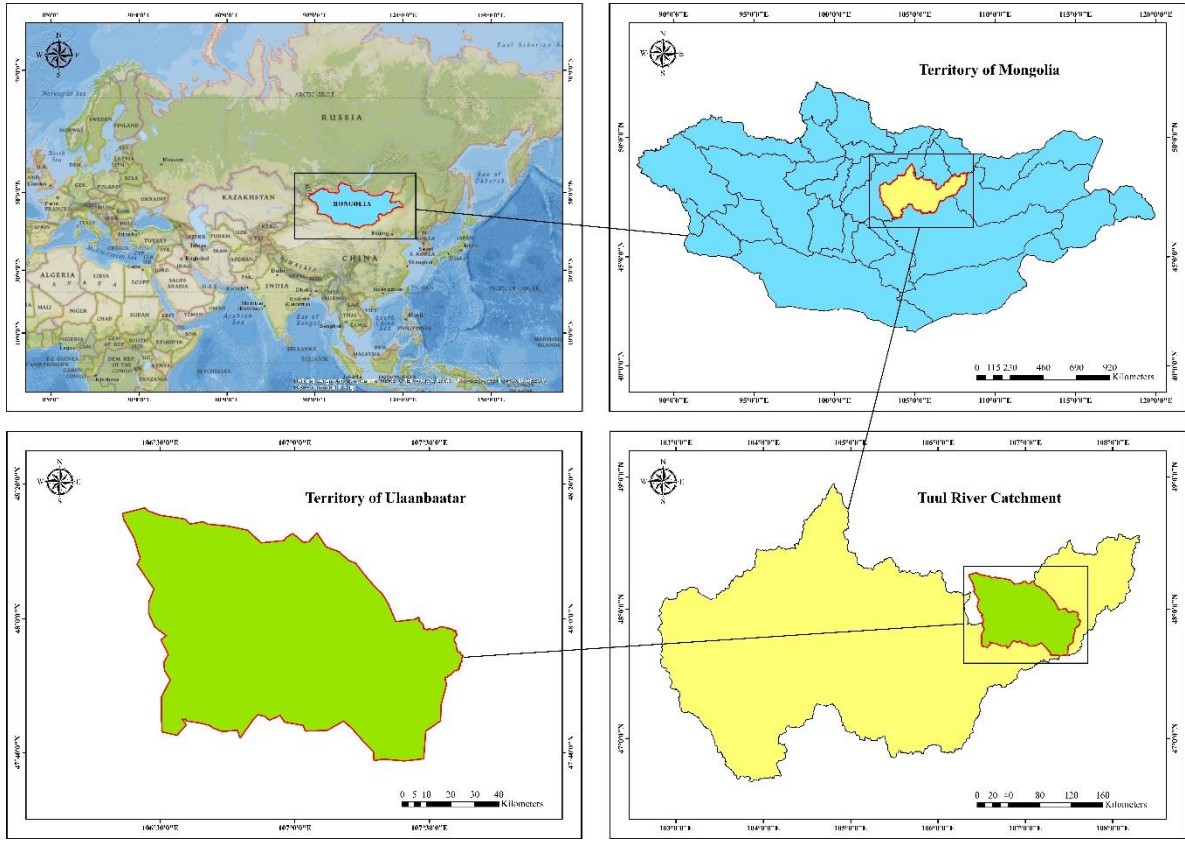

**Figure 1.** Study area of Ulaanbaatar city, which is the capital of Mongolia.

According to the Mongolian law on water, all water basin areas should have IWRM plans and update these once every five years. The territory of Ulaanbaatar city is 4704 square km, which occupies 8% of the basin [28]. However, the population in the city occupies 96% of the total population in the basin and 46% of the total population of the country [29]. Recently, water management of the capital has been included in the basin management plan. IWRM is a relatively young subject in Mongolia, and especially Urban Water Management is particularly new.

## 3. Results

### 3.1. Results of the Trends and Pressures Framework (TPF) Analysis

The results of the TPF analysis for Mongolia are shown in Figure 2. None of the indicators were assessed as 'no concern'. The number of 'low concern' indicators is two, whereas seven of the total indicators were scored as 'medium concern', one was scored as 'concern', while two indicators were scored as 'great concern'.

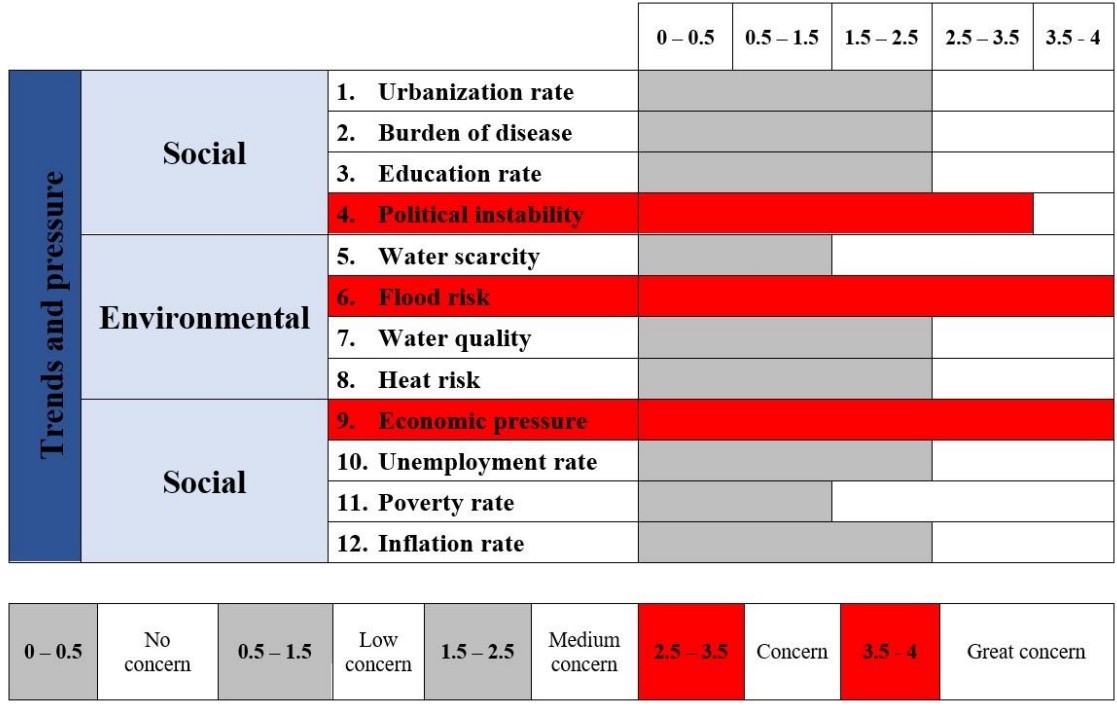

**Figure 2.** Results of the Trends and Pressures Framework analysis of Mongolia and Ulaanbaatar.

Water scarcity is currently a 'low concern' but may become a low or medium concern as a result of increasing future water demand for industrial and domestic use [30,31]. Recent literature has shown that water scarcity is socially constructed. It is created by people and institutions for a variety of reasons, often as a result of poor water governance. Moreover, water scarcity is a great concern globally due to economic growth, changing consumption patterns and a variety of environmental issues [32–34]. Heat risk was assessed as one of the 'medium concern' indicators. As observed in research on the Urban Heat Island (UHI) in Ulaanbaatar, it was shown that the UHI depends on seasonal variation. UHI intensity is weakest in summer and strongest in winter due to burning of organic materials, coal and wood [35]. Water quality was assessed as 'medium concern', and this has been indicated by previous research on water contamination in Ulaanbaatar. Pollution of the water wells in Ulaanbaatar was observed based on chemical analyses [36]. Flood risk and economic pressure are a 'great concern'. Economic pressure is based on the scoring of the Gross Domestic Product (GDP) per capita of Mongolia, and reflects the economic power of a country. The low GDP per capita for Mongolia may imply that there is limited capacity to fund many activities or programs related to water management, water governance and water infrastructure. Moreover, the economic pressure is confirmed by indicator 17, i.e., the operating costs recovery (ratio), showing that the operational revenue is less than the operational cost. As reported by local newsletters and reports, Ulaanbaatar faces a high risk of flooding during heavy rain due to the fact that there are many paved areas, leading to a high percentage of soil sealing, which limits water retention. This is confirmed by the high soil sealing of Ulaanbaatar, which is 70% (indicator 6.1).

*3.2. Results of the City Blueprint Framework (CBF) Analysis*

The scores of the 25 indicators of the CBF are shown in Figure 3. The BCI of Ulaanbaatar is 2.3, which categorizes Ulaanbaatar as a Wasteful City (Table 3). This means that basic water services are largely met, but flood risk can be high and Waste Water Treatment (WWT) is poorly covered.

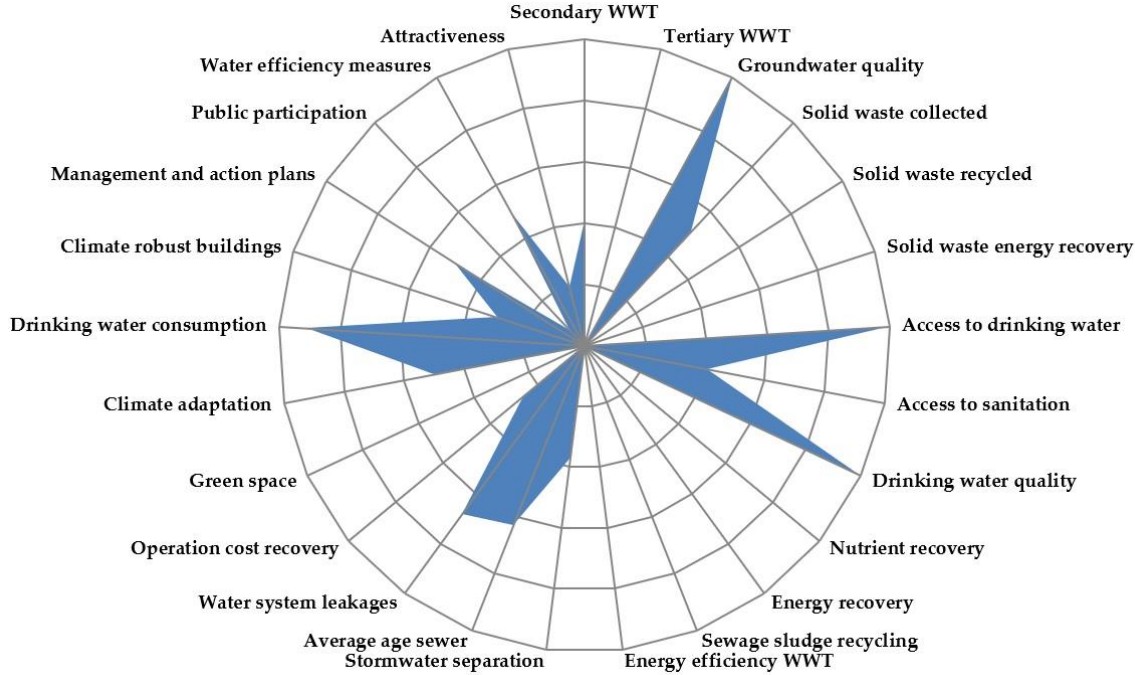

**Figure 3.** City Blueprint Framework results for Ulaanbaatar.

The spider diagram clearly illustrates the low performance of Waste Water Treatment (WWT) in general. In fact, all indicators related to WWT (indicators 1, 2, 10, 11, 12 and 13) are scored with 0 or nearly 0 points. Often, only primary and a small portion of secondary WWT is applied, leading to large scale pollution. Water consumption and infrastructure leakages are high due to insufficient infrastructure maintenance. Solid waste production is high and almost all waste is dumped in landfills, which contributes to water pollution. Furthermore, indicators of climate robustness are scored low (indicators 18, 19, 21), which indicates that climate adaptation actions are mostly deficient. The operational costs are higher than the operational revenues for water supply and sanitation services. Some of the stormwater sewers are connected to sanitary sewers, and during periods of heavy rain, sanitary systems are clearly affected. The attractiveness of water bodies in Ulaanbaatar is limited because of pollution. Moreover, blue infrastructures in the city such as fountains and ponds are very few, and a considerable number of these blue infrastructures are actually out of service. The most attractive natural water body is the Tuul River, and prices of residential properties next to this river are higher than other locations. The indicators with the highest scores are drinking water quality, access to drinking water, and drinking water consumption. The water supply and sewerage authority regularly takes groundwater monitoring samples and analyzes physical and chemical parameters. Based on the results of this monitoring program, all samples met the drinking water standards in Mongolia.

### 3.2. Results of the Governance Capacity Framework (GCF) Analysis

The analysis revealed a total of five indicators out of the 27 GCF indicators with a limiting (-) score. Furthermore, five indicators show encouraging (+) scores, and another 17 indicators (68%) are scored as indifferent (0) to the overall water governance capacity. This shows that there is substantial room to improve both water management (IWRM; Figure 3) and water governance in Ulaanbaatar (Table 5 and Figure 4).

**Table 5.** Governance Capacity Framework results for Ulaanbaatar on IWRM.

| Indicators | Scale | Description |
|---|:---:|---|
| 1.1: Community knowledge | 0 | Underestimation |
| 1.2: Local sense of urgency | 0 | Sense of urgency of long-term sustainability goals |
| 1.3: Behavioral internalization | + | Moderate internalization |
| 2.1: Information availability | 0 | Information fits demand, limited exploratory research |
| 2.2: Information transparency | + | Sharing of partly cohesive knowledge |
| 2.3: Knowledge cohesion | 0 | Insufficient cohesion between sectors |
| 3.1: Smart monitoring | − | Reliable data but limited coverage |
| 3.2: Evaluation | − | Non-directional evaluation |
| 3.3: Cross-stakeholder learning | + | Open for cross-stakeholder learning |
| 4.1: Stakeholder inclusiveness | 0 | Untimely consultation and low influence |
| 4.2: Protection of core values | 0 | Suboptimal protection of core values |
| 4.3: Progress and variety of options | 0 | Consultation or short active involvement |
| 5.1: Ambitious and realistic management | + | Long-term ambitious goals |
| 5.2: Discourse embedding | 0 | Low sense of urgency embedded in policy |
| 5.3: Management cohesion | 0 | Fragmented policies |
| 6.1: Entrepreneurial | 0 | Conventional and risk-averse entrepreneurship |
| 6.2: Collaborative | 0 | Agents enhance conventional collaboration |
| 6.3: Visionary | − | Unilateral and short-term vision |
| 7.1: Room to maneuver | 0 | Limited room for innovation and collaboration |
| 7.2: Clear division of responsibilities | − | Barriers for effective cooperation |
| 7.3: Authority | 0 | Restricted authority |
| 8.1: Affordability | 0 | Unaffordable climate adaptation |
| 8.2: Consumer willingness to pay | + | Willingness to pay for provisional adaptation |
| 8.3: Financial continuation | 0 | Financial continuation for basic services |
| 9.1: Policy instruments | 0 | Fragmented instrumental use |
| 9.2: Statutory compliance | − | Moderate compliance to incomplete legislation |
| 9.3: Preparedness | 0 | Low awareness of preparation strategies |

As shown in Figure 4, indicator 7.2 is scaled as limiting (−). In Ulaanbaatar, responsibilities related to IWRM are fragmented. They are divided over a number of government organizations, as explained in the Introduction. This fragmentation creates uncertainty because of overlapping responsibilities. Furthermore, indicator 1.1 is scaled as indifferent (0). This illustrates that most communities have a basic understanding of water challenges. However, the actual risks, impacts and frequencies are often not fully known and are generally underestimated. Future risks, impacts and frequencies are often unknown. Some awareness has been raised among or is created by local stakeholders and communities. Moreover, indicator 8.2 is scored as encouraging (+) since the polluter-pays principle has been introduced. Due to inexperience, implementation of this principle is however often suboptimal.

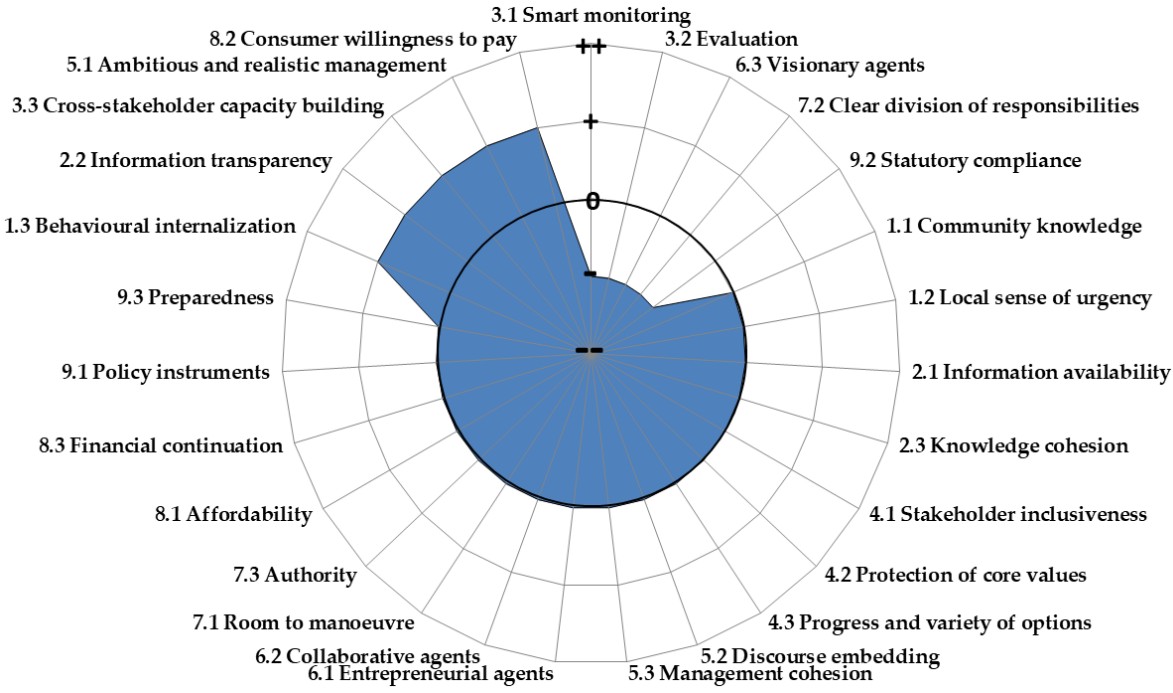

**Figure 4.** Governance Capacity Framework results for Ulaanbaatar on IWRM.

## 4. Discussion

The results of this research show that there is strong demand to develop an independent IWRM plan for the capital Ulaanbaatar. This is also needed for IWRM in the Tuul basin area. The reasons are obvious: almost half of the population (46%) of Mongolia lives in Ulaanbaatar, the capital, and the city area occupies only 8% of the Tuul basin area. The CBA can be an adequate assessment tool to facilitate the implementation of IWRM at country, river basin and city level. In order to coordinate the IWRM plans at different levels, urban IWRM plans must comply with IWRM at national and river basin levels. Plans for IWRM in cities can be more focused and be more practical, taking into consideration the goals set at the other IWRM levels.

As of 2018, 11 cities in Asia (Jakarta, Manila, Ahmedabad, Bandung, Tianjin, Ho Chi Minh City (HCMC), Bangkok, Hohhot, Taipei, Seoul and Singapore) had been assessed by the City Blueprint Approach [20]. Based on the BCI, one city (Jakarta) was categorized as "Cities lacking basic water services", five cities (Ahmedabad, Bandung, Bangkok, HCMC and Manila) were categorized as "Wasteful cities", three cities (Hohhot, Taipei, Tianjin) were categorized as "Water efficient cities", and two cities (Seoul, Singapore) were "Resource efficient cities" [14]. The cities with high BCI values, i.e., Seoul and Singapore, also have the highest GDP in comparison with the other assessed cities [22]. Approximately half of these Asian cities have GDPs between 2500 and 7000 USD and are in the same Blue City Index (BCI) categorization as Ulaanbaatar, which has a GDP of 4100 USD [26].

Since 2011, the CBA has been modified following a learning-by-doing approach. The current version is based on the review of Koop published in 2015 [13,14], and the last update was provided in 2017 when the GCF was developed and implemented [21] as part of the CBA. The CBA has been applied most frequently in Europe. Most European countries are coastal countries. This may have its limitations for the use of the CBA for landlocked countries outside Europe. For instance, minimum and maximum values of the indicators 4—Solid waste collected, 15—Average age of sewer, 18—Green space and 20—Drinking water consumption use European data, which may be different in developing countries. Furthermore, there is some difference between the settled (European) and nomadic (Asian) lifestyle in terms of water supply and sewerage. In Asian nomadic lifestyles, such as those of dwellers in the Ger area in Ulaanbaatar, Mongolia and Central Asian countries, there are still residents who are carrying drinking water away from their homes and are using unsafe pots or tanks. In addition, Mongolians are not connected to a central sewerage system and may still be using

pit latrines [37]. As of 2017, 69.3% of the urban population in Landlocked Developing countries has safely managed drinking water, and 60.7% has basic sanitation services [38]. These aspects must be taken into consideration at the political level to prioritize actions to improve IWRM at country, river basin and city level in Mongolia. It may also imply the development of a modified version of the CBA for landlocked countries such as Mongolia, Afghanistan, Laos, Nepal, Kyrgyzstan, etc. We conclude that the next most populated centers in Mongolia are Darkhan and Erdenet cities, for which we recommend the expansion of this study to these two cities because of the importance of IWRM and SDG6 for the general well-being of the people of Mongolia [34,39].

Operation cost recovery is one of the indicators with a low score. Total annual operational revenues of drinking water and the sewerage system are lower than the total annual operating costs. To improve water management in Ulaanbaatar, projects financed with resources from international institutions can be implemented. For example, the project 'Strengthening water resources management in Mongolia' was implemented between 2007 and 2012 and received funding from the Government of the Netherlands. Also, the project 'Re-creation of the Waste Water Treatment Plant in Ulaanbaatar' was funded by a loan from the Government of China. Finally, the project 'Darkhan Waste Water Management' in Darkhan city, the next most densely populated area in Mongolia, received funding from the Asian Development Bank.

General information about water is open whereas information about water challenges among the general public is actually rather limited. According to the Mongolian law on water, public participation in discussions of IWRM at river basin level is an open participative process. The river basin authorities develop annual reports and submit these to the Ministry of Environment, but actually the reports are not fully accessible to the public. The Ministry of Environment and the Water Supply and Sewerage Authority in Ulaanbaatar provide posters or graphical illustrations about water resources and consumption for the general public. However, community knowledge, one of the GCF indicators scaled as indifferent, shows that the actions mentioned above can be insufficient. Moreover, visionary agents and a clear division of responsibilities are scaled as limiting as the responsibility for water issues is shared among different organizations, as mentioned in the Introduction. The Ministry of Water was established in 1965 and operated until 1986. Then, the Water Authority was established in 2005 and was overthrown in 2012. Therefore, the re-establishment of a central governmental organization for IWRM is needed at national, river basin and city level.

Water is a global challenge, and best practices and solutions have been found [15]. UNESCO's Urban Water Management Programme is proposed to help countries by promoting science-based policy, scientific knowledge and information on new and innovative approaches, solutions and tools for sustainable urban water management, as well as by providing capacity building support [40]. Long-term planning is crucial [21,34]. For instance, urban water management plans are developed by water suppliers in California, USA every five years. These plans support the suppliers' long-term resource planning to ensure that adequate water supply is available to meet existing and future water needs. The California Department of Water Resources is responsible for these plans [41]. For waste water treatment, Singapore's example is one of the world's best practices. The NEWater program uses waste water treatment technology with high efficiency. The treated water has been tested with a lot of scientific tests and meets international standards. The institution responsible for this program is Singapore's National Water Agency. This makes Singapore one of the leading cities, with a very high BCI [20]. These examples can be implemented in Ulaanbaatar to improve water management and governance.

## 5. Conclusions

This study focused on IWRM in Mongolia, and in particular on urban water management in Ulaanbaatar city. Based on the results of a broad diagnosis using the City Blueprint Framework, the Trends and Pressure Framework and the Governance Capacity Framework, the following conclusions can be drawn:

1. The risk of flooding and economic pressure have a great impact on the water sector in Ulaanbaatar city.

2.  The Blue City Index of Ulaanbaatar is 2.3 (out of 10), which categorizes Ulaanbaatar as a wasteful city. WWT is the highest priority in terms of water management improvement. All indicators related to waste water treatment (indicators 1—Secondary waste water treatment, 2—Tertiary waste water treatment, 10—Nutrient recovery, 11—Energy recovery, 12—Sewage sludge recycling and 13—Energy efficiency waste water treatment) are scored with 0 or nearly 0 points. Often, only primary and a small portion of secondary WWT is applied, leading to large-scale pollution. The treatment percentage of the Central Waste Water Treatment Plant of Ulaanbaatar is only 50%–60% and waste water is directly discharged into the Tuul river [42], and this shows that waste water from the Central Waste Water Treatment Plant of Ulaanbaatar is the main source of pollution of the Tuul river. Both waste water and sewage sludge cause odor pollution in surrounding areas during summer time. Furthermore, the low green space area of Ulaanbaatar increases flood risk during heavy rain and causes huge surface runoff, sometimes leading to the destruction of roads and houses. These situations increase the risk of damage to nature, human health and infrastructure.

3.  Water consumption and infrastructure leakages are high due to the lack of environmental awareness and infrastructure maintenance. Operation cost recovery is not enough to sustain urban water services in Ulaanbaatar.

4.  During the process of collecting information, a lack of open-source information was observed, and some data were missing. Therefore, improvement in data management and transparency is needed.

5.  Water governance in Ulaanbaatar is not sufficient. Monitoring, evaluation, institutional fragmentation and statutory compliance are among the factors that need to be addressed.

**Supplementary Materials:** The following are available online at www.mdpi.com/xxx/s1, Table SA1: Scoring for Burden of disease by Disability-Adjusted Life Year (DALY), Table SA2: Scoring for Fresh water scarcity by percentage of renewable fresh water resource abstracted, Table SA3: Scoring for Groundwater scarcity by percentage of abstracted renewable ground water recharge, Table SA4: Scoring for Sea water intrusion, Table SA5: Scoring for Groundwater salinization, Table SA6: Scoring for Sea level rise by percentage of urban affected area with sea water, Table SA7: Scoring for Sea level rise by percentage of urban affected area with river water, Table SA8: Scoring for Flood risk due to subsidence, Table SB1: A self-assessment meaning with corresponding indicator score.

**Author Contributions:** Methodology, S.K., K.v.L.; data collection, E.M., T.B.; analysis, E.M.; investigation, S.K., A.O.; writing—original draft preparation, E.M.; writing—review and editing, A.O., S.K., K.v.L.; supervision, A.O., K.v.L.; project administration, A.O.; funding acquisition, A.O.

**Funding:** This research was funded by National University of Mongolia, grant number P2018-3594.

**Acknowledgments:** The authors would like to thank the interviewees involved in the Governance Capacity Framework assessment who have given us the most valuable information, Davaa Gombo and Basandorj Davaa, the Water Supply and Sewerage Authority in Ulaanbaatar, the Information and Research Institute of Meteorology, Hydrology and Environment, the Mongolian University of Science and Technology, and the Ministry of Environment and Tourism.

**Conflicts of Interest:** The authors declare no conflict of interest.

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
