# Peer review of "Application of the City Blueprint Approach in Landlocked Asian Countries: A Case Study of Ulaanbaatar, Mongolia"

_water, doi:10.3390/w12010199_

Round 1

Reviewer 1 Report

This document applies the City Blueprint Approach (CBA), a diagnosis tool to assess IWRM of Ulaanbaatar city, Mongolia.

This methodology is already sufficiently contrasted, it has been applied in cities of developed and developing countries such as Rotterdam, Dar es Salaam (Tanzania), Istanbul, Hamburg, Amsterdam, Ho Chi Minh City (Vietnam), Melbourne, Quito (Ecuador ) and Ahmedabad (India). It has also been applied in the case of Cape Town (Madonsela, B., Koop, S., Van Leeuwen, K., & Carden, K. 2019) file: /// C: / Users / UJA / Downloads / water- 11-00292-v2% 20 (2) .pdf

The work is well structured, the analysis developed is solvent. The authors present the method and the materials used, the results of the research and their discussion to, finally, refer the main conclusions obtained.

The paper describes the situation presented by the city of Ulaanbaatar in relation to Integrated Water Resources Management (IWRM). I believe that the authors should provide more arguments that take into account the level of development of the country and, also, of an institutional nature that explain, to some extent, the reality analyzed. This additional information would allow answering questions such as the following: Does this country have any specific regulations that determine its obligations to improve water resources management? Is there a standard similar to EU Directive 91/271 / EEC - urban wastewater treatment? To what extent are actions being implemented to raise public awareness of the challenges of water management? Could projects for the improvement of water management be financed with resources from international institutions such as the World Bank or the European Development Fund? What actions have been implemented in other countries and could be applied in Ulaabaatar to improve the integrated management of water resources?

What effects can the population and the environment have that The Blue City Index of Ulaanbaatar is 2.3 points (out of 10), which categorizes Ulaanbaatar as wasteful city?

However, the work is very interesting.

Author Response

Thank you so much for your valuable recommendation and comments on our manuscript.

Reviewer 2 Report

Overall, a well written, structured, paper. Attached I have put some intext comments on how to improve certain sentences and some additional relevant references that could be added.

Author Response

Thank you so much for your valuable recommendation. 

A sentence in line 50 is rephrased.

We added references that you recommended to us. 

Reviewer 3 Report

This is an interesting case study. The results are clearly presented. Just some comments:

What exactly is an IWRM approach in the context of the case study? Were the interviewees only asked about the current perception on trends and pressure (but not future)? Line 24 “water waste water” wastewater? Line 50 grammar mistake Lines 268-269 “The results of this research show that there is strong demand to develop an independent IWRM plan for the capital Ulaanbaatar.” In that case, would there be conflicting actions from the IWRM plans of the capital city, the catchment area and the country? While there are results for the governance aspects, a discussion on them is lacking.

Author Response

Thank you so much for your valuable recommendation and comments.

Reviewer 4 Report

The manuscript presented a very interesting topic, which needs an attention of the whole world. Also, the paramteres considered in the study are well appreciated. But, the overall study is not sufficient to prove author's point. It is not clear how the conclusions are derived from the defined input parameters. It is recommended to authors to provide a detailed quantative analysis prior to concluding the outcomes. Also, outcomes and results in tabular format can ne helpful for the manuscript. Further the presentation of manuscript is well described. Very few grammatical mistakes are observed.

Author Response

(The authors gave the same response as above.)

Round 2

Reviewer 2 Report

While it is improved, my only pending comment is on the substantiation of line 50 on SDG6; I would suggest in providing stronger substantiation by inclusion of the different suggestions I made during the first round of review, as mentioned:

Please add reference to the latest literature on SDG6 published in MDPI:
https://www.mdpi.com/2073-4441/9/6/438
https://www.mdpi.com/2071-1050/9/9/1572/htm
https://www.mdpi.com/2071-1050/10/10/3640

Apart from this, it seems much stronger than it was.

Author Response

Thank so much again for your valuable advice.

We added references according to your suggestion on SDG 6. 

Reviewer 3 Report

I am generally satisfied with the revision. However, please carefully proofread the paper, especially the revised parts. It contains numerous errors. Just list some of them here, many more....

Line 314 "one of the low scored indicators"?

Line 320, 321 "has been funded by"?

Line 321 "the next densely populated area"?

Line 320 "more sensitive to flooding"?

Line 371 "are increasing risk to"?

---

Lines 336-348 are not so relevant. 

Author Response

Thank you so much again for your advice.

The manuscript carefully edited for the English language by a native speaker and scientist who has experience in paper writing.

Reviewer 4 Report

The response from reviewer are satisfactory. I would like to recommend the manuscript for acceptance. 

Author Response

Thank you so much for your valuable revision and cooperation.